# Multimodal Few-Shot Learning with Frozen Language Models

**Maria Tsimpoukelli**[*]
DeepMind
mrts@deepmind.com

**Jacob Menick**[*]
DeepMind
University College London
jmenick@deepmind.com

**Serkan Cabi**[*]
DeepMind
cabi@deepmind.com

**S. M. Ali Eslami**
DeepMind
aeslami@deepmind.com

**Oriol Vinyals**
DeepMind
vinyals@deepmind.com

**Felix Hill**
DeepMind
felixhill@deepmind.com

## Abstract

When trained at sufficient scale, auto-regressive language models exhibit the notable ability to learn a new language task after being prompted with just a few examples. Here, we present a simple, yet effective, approach for transferring this few-shot learning ability to a multimodal setting (vision and language). Using aligned image and caption data, we train a vision encoder to represent each image as a sequence of continuous embeddings, such that a pre-trained, frozen language model prompted with this prefix generates the appropriate caption. The resulting system is a multimodal few-shot learner, with the surprising ability to learn a variety of new tasks when conditioned on examples, represented as a sequence of multiple interleaved image and text embeddings. We demonstrate that it can rapidly learn words for new objects and novel visual categories, do visual question-answering with only a handful of examples, and make use of outside knowledge, by measuring a single model on a variety of established and new benchmarks.

## 1 Introduction

Auto-regressive transformers have been shown to be very impressive models of natural language [42]. Large-scale language transformers exhibit several surprising abilities beyond that of standard text generation [4, 31]. Perhaps most notably, they are *few-shot learners*; they can learn to perform a new task from a few examples without any further gradient updates. Equipped with this ability, these models have been shown to rapidly adapt to new tasks and styles of generation via prompting (e.g. switching from formal to informal language) [4], to retrieve relevant encyclopedic or general knowledge when primed with a relevant context (e.g. answering questions such as 'When did the French Revolution begin?') [34, 1, 28] and to use new words in appropriate ways straight after being taught what those words mean (sometimes referred to as 'fast binding') [12, 4].

Despite these impressive capabilities, such large scale language models are 'blind' to modalities other than text, preventing us from communicating visual tasks, questions or concepts to them. Indeed, philosophers and linguists have questioned whether an un-grounded language model can ever achieve true understanding of the language it processes [5, 2]. Here, we present *Frozen*, a method for giving a pre-trained language model access to visual information in a way that extends its few-shot learning capabilities to a multimodal setting, without changing its weights. *Frozen* consists of a neural network trained to encode images into the word embedding space of a large pre-trained language model such that the language model generates captions for those images. The weights of the language model are kept frozen, but gradients are back-propagated *through* it to train the image encoder from

35th Conference on Neural Information Processing Systems (NeurIPS 2021).

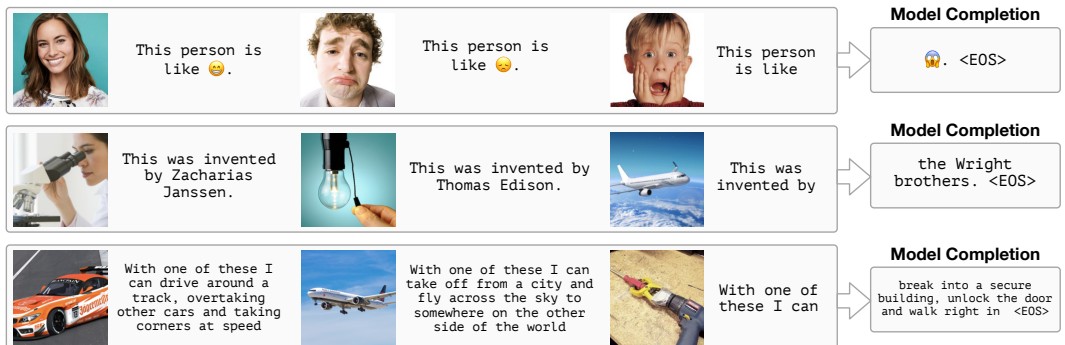

Figure 1: Curated samples with about five seeds required to get past well-known language model failure modes of either repeating text for the prompt or emitting text that does not pertain to the image. These samples demonstrate the ability to generate open-ended outputs that adapt to both images and text, and to make use of facts that it has learned during language-only pre-training.

scratch (Figure 2). Although *Frozen* is trained on single image-text pairs, once trained it can respond effectively to interleaved sequences of multiple images and text. This allows users to 'prompt' it with several examples of new multimodal tasks before evaluating its performance, or to 'teach' it the name of a new visual category before immediately asking about that category.

By exploiting its pre-trained language model, *Frozen* exhibits nontrivial zero-shot performance on multimodal tasks that it was not trained on, such as visual question answering (VQA). More surprisingly, it gets better at these tasks after seeing a handful of examples 'in-context' as in [4], and also performs above chance on tests of fast category learning such as miniImageNet [43]. In each case, comparisons with 'blind' baselines show that the model is adapting not only to the language distribution of these new tasks, but also to the relationship between language and images. *Frozen* is therefore a *multimodal few-shot learner*, bringing the aforementioned language-only capabilities of rapid task adaptation, encyclopedic knowledge and fast category binding to a multimodal setting.

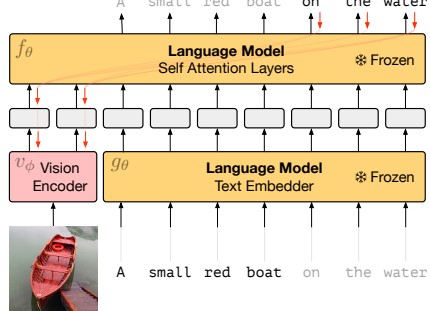

Figure 2: Gradients through a frozen language model's self attention layers are used to train the vision encoder.

Our goal in developing *Frozen* was not to maximise performance on any specific task, and in many cases it is far from state-of-the-art. Nonetheless, it performs well above trivial baselines across a wide range of tasks without ever seeing more than a handful of the training examples provided by these benchmarks. Moreover, as illustrated in Figure 1, *Frozen* is a system for genuinely open-ended and unconstrained linguistic interpretation of images that often produces compelling output.

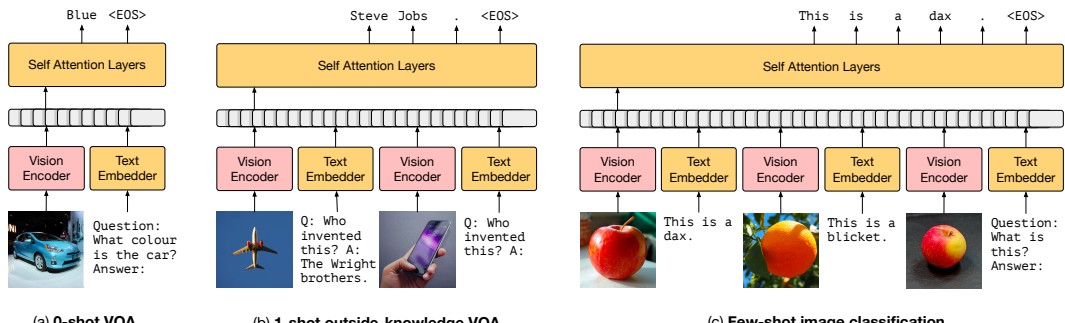

Figure 3: Inference-Time interface for *Frozen*. The figure demonstrates how we can support (a) visual question answering, (b) outside-knowledge question answering and (c) few-shot image classification via in-context learning.

To summarise, our contributions are as follows: 1. We present *Frozen*, a modular, scalable and efficient approach to training vision front-ends for large language models. The resulting combined model retains all of the capabilities of large language models, but can also process text and image inputs in any arbitrary sequence. 2. We show that such models transfer their capacity for rapid task adaptation, encyclopedic knowledge and fast category binding from a language-only to a multimodal setting, and verify that prompting them with both visual and language information can be strictly more effective than doing so with language information alone. 3. We quantify these capabilities on a range of existing and new benchmarks, paving the way for future analysis of these capabilities.

## 2 Related Work

The *Frozen* method is inspired by lots of recent work. [26] show that the knowledge encoded in transformer language models can be a valuable prior for tasks involving reasoning and memory across discrete sequences, and even classification of images presented as sequences of spatial regions. In that approach, a small subset of the pre-trained language model weights are fine-tuned to the various final applications. In contrast, applying *Frozen* to different tasks does not involve any weight updates to the transformer whatsoever; the system adapts to and improves at multimodal (vision and language) tasks as activations propagate through the model. The two studies thus reveal different ways in which knowledge acquired from text can transfer to non-linguistic settings.

The effectiveness of *prefix tuning* [23] or *prompt tuning* [20] was another important motivation for *Frozen*. Prefix tuning is a method for prompting a language model to produce output of a particular style using gradient descent to learn a task-specific bias term which functions like the continuous embedding of a text prompt. Using prefix tuning, language models can be adapted to different natural language generation tasks like summarization. *Frozen* could also be considered a type of *image-conditional prefix tuning*, in which this continuous prefix is not a bias but an *image-conditional* activation produced by an external neural network.

Learning to embed image representations into the 'word' space of a large pretrained language model was done previously by [16]. This work focused on image-text classification, and uses a BERT-style language model that is fine-tuned (rather than frozen) on multimodal data. [36] extend a similar image embedding+BERT system to create a generative model of text, using a pre-trained object extraction system to embed images into word space. Neither of these studies consider the problem of learning image-text correspondences in a few shots.

A large body of work has applied either text-specific or multimodal representation-learning approaches like BERT [8] to visual question answering (VQA) and captioning (see e.g. [25, 40] and many more). In these approaches, models are first trained with aligned data on task-agnostic cross-modal objectives and then fine-tuned to specific tasks. This approach can yield state-of-the-art performance on a range of classification tasks. Unlike *Frozen*, the resulting systems are highly specialized to one task, and cannot learn new categories or adapt to new tasks in a few shots.

By contrast, [7] propose text generation as an objective for task-general multimodal models, yielding a system that, like *Frozen*, produces unconstrained language output. Unlike *Frozen*, they do not use a pre-trained model trained on text only, and do not consider zero or few-shot learning, instead updating all weights of the system with training data for each task they consider – thus, again, specializing the models to one task at a time. Similarly, [46] and [6] show that a large pre-trained language model as decoder can improve a captioning performance when training data is limited. Unlike *Frozen*, they use pre-trained frozen visual encoders or object extractors and fine-tune the pre-trained weights in the text decoder on the captioning data. Similarly, they do not consider zero or few-shot adaptation across different multimodal tasks. Past work has also explored alternative approaches for post-hoc combination of models for different modalities using latent variables [41].

Multimodal pre-training has recently been shown to enable strong zero-shot generalization in the discriminative setting using large-scale contrastive learning [29, 14]. Also in a discriminative setting, [45] has observed signs of emergent few-shot-learning from large-scale training. In contrast, our work enables strong generalization to new multimodal tasks both zero-shot or few-shot with completely open-ended generative text output.

# 3 The *Frozen* Method

*Frozen* is a method for grounding a large language model without changing its weights, closely related to *prefix tuning* [23, 20]. Prefix tuning trains a task-specific continuous bias term to function like the embedding of a constant, static text prompt used for all test-time examples. *Frozen* extends this approach by making this prefix *dynamic*, in that it is not a constant bias but an input-conditional *activation* emitted by a neural network.

## 3.1 Architecture

**Pre-trained Autoregressive Language Models**   Our method starts from a pre-trained deep auto-regressive language model, based on the Transformer architecture [42, 30], which parametrizes a probability distribution over text $\mathbf{y}$. Text is decomposed into a sequence of discrete tokens $\mathbf{y} = y_1, y_2, ..., y_L$ by the SentencePiece tokenizer [18]. We use a vocabulary of size 32,000. The language model makes use of an embedding function $g_\theta$ which independently transforms each token into a continuous embedding $t_l := g_\theta(y_l)$, as well as a transformer neural network $f_\theta$ whose output is a vector of logits parameterizing a categorical distribution over the vocabulary. The distribution $p_\theta(\mathbf{y})$ is represented as follows:

$$\log p_\theta(\mathbf{y}) = \sum_l \log p_\theta(y_l|y_1, y_2, ..., y_{l-1}) = \sum_l f_\theta(t_1, t_2, ..., t_{l-1})_{y_l}$$

The model we start from is pre-trained, i.e. $\theta$ has been optimised via the standard maximum-likelihood objective on a large dataset of text from the internet. We use a 7 billion parameter transformer trained on the public dataset C4 [31] – previous work has shown that the multi-billion parameter scale is sufficient to exhibit the key capacities we are interested in studying [30, 34].

**Vision Encoder**   Our vision encoder is based on NF-ResNet-50 [3]. We define $v_\phi$ as a function that takes a raw image and emits a continuous sequence to be consumed by the transformer. We use the final output vector of the NF-Resnet *after* the global pooling layer.

**Visual Prefix**   One important requirement is to represent images in a form that the transformer *already* understands: a sequence of continuous embeddings, each having the same dimensionality $D$ as a token embedding $t_l$. We therefore form the visual prefix by linearly mapping the vision encoder's output to $D * k$ channels, and then reshaping the result as a sequence of $k$ embeddings, each with dimensionality $D$. We call this sequence a *visual prefix* since it plays the same functional role in the transformer architecture as (part of) an embedding sequence of prefix tokens. We experimented using different number of tokens $k$, specifically 1, 2 and 4 and found that 2 performs best, though certainly this would be sensitive to other architectural details. See Appendix for more details on the architecture.

## 3.2 Training

During training, we update only the parameters $\phi$ of the vision encoder using paired image-caption data from the Conceptual Captions dataset [37]. Our experiments show that fine-tuning $\theta$ hurts generalization, as much less paired image-caption data is available than the amount of text-only data used to pre-train $\theta$. Training only the parameters $\phi$ makes our system *modular* – it can use an existing language model off the shelf – and also quite simple: we only train a visual encoder and rely on the capabilities of an existing language model.

Following standard captioning systems [22, 13], we treat captioning as conditional generation of caption text $\mathbf{y}$ given an image $\mathbf{x}$. We represent $\mathbf{x}$ as $v_\phi(\mathbf{x}) = i_1, i_2, ..., i_n$ and train $\phi$ to maximise the likelihood:

$$\log p_{\theta,\phi}(\mathbf{y}|x) = \sum_l \log p_{\theta,\phi}(y_l|\mathbf{x}, y_1, y_2, ..., y_{l-1})$$
$$= \sum_l f_\theta(i_1, i_2, ..., i_n, t_1, t_2, ..., t_{l-1})_{y_l}$$

Whilst the parameters $\theta$ are frozen, each element $i_k$ of the visual prefix receives gradients $\sum_l \nabla_{i_k} f_\theta(i_1, i_2, ..., i_n, t_1, t_2, ..., t_{l-1})_{y_l}$, enabling the parameters of the visual encoder to be optimised with standard backpropagation and SGD (Figure 2).

As the notation $f_\theta(i_1, i_2, ..., i_n, t_1, t_2, ..., t_{l-1})$ suggests, we present the visual prefix during training as if it were a sequence of embeddings occurring earlier in time than the caption (token embeddings) $t_1, t_2, ...$. We use relative positional encoding [38], which enables the transformer to generalize to prefix sequences where an image is not always in the first absolute positions, and where more than one image may be present. In particular, we use the version of relative attention described in transformerxlDai.We leave improvements of this simple scheme for future work.

### 3.3 Interface at Inference Time

At inference time, a vanilla language model, conditioned upon an arbitrary text prompt $y_1, y_2, ..., y_p$, generates text sequences $y_{p+1}, y_{p+2}, ...$ autoregressively. In *Frozen* it is straightforward to include images in such prompt by placing an image's embedding $i_1, i_2$ as a prefix to a text embedding subsequence $t_1, t_2, ..., t_p$. Because the transformer $f_\theta$ is modality-agnostic, we can interleave a sub-sequence of text token embeddings with a sub-sequence of image embeddings in any arbitrary order. In Figure 3, we show how this can support zero-shot visual question-answering (Figure 3a), few-shot visual question-answering (Figure 3b), and few-shot image classification (Figure 3c).

To evaluate these tasks, the model *decodes* output sequences greedily and these outputs are compared against the ground truth answers of the task following the normalization technique used in [19]. To probe the open-ended capabilities of *Frozen*, we decided not to use common practice of short-lists of pre-canned answers, even though in some tasks this may hurt its performance in accuracy percentages.

### 3.4 Few-Shot Learning Definitions

The ability of *Frozen* to be conditioned on a sequence of interleaved images and text allows it not only to be able to perform different multimodal tasks, but also gives rise to different ways of 'inducing' the task to the model in order to improve its performance. We briefly define the terminology used in our settings, common amongst all the different tasks. See Figure 5 in the appendix for a visual illustration of these concepts.

- **Task induction** Explanatory text that precedes the sequence of images and text. It is intended to describe the task to the model in natural language, for example 'Please answer the question.'
- **Number of shots** The number of distinct full examples of the task presented to the model prior to the evaluated example. For example, in Visual Question-Answering, a shot is an image along with the question and the answer.

For tasks involving fast category binding (e.g., few-shot image classification), we define further specific terminology. See also Figure 4a and Figure 6 in the appendix.

- **Number of ways** The number of object classes in the task (e.g. dog vs cat).
- **Number of inner-shots** The number of distinct exemplars from each category that are presented to the model (i.e. number of images of different dogs). In previous work with MiniImagenet, these were known as *shots*, but we modify the term here to distinguish from the more general usage of the term described above.
- **Number of repeats** The number of times each inner-shot is repeated in the context presented to the model. We use this setting as an ablation to explore how the model integrates visual information about a category.

## 4 Experiments: A Multi-Modal Few-Shot Learner

Our experiments are designed to quantify three capacities that should be characteristic of a Multi-Modal Few-Shot Learner: *rapid adaptation* to new tasks, fast access to *general knowledge* and *fast binding* of visual and linguistic elements. We train *Frozen* on Conceptual Captions, a public dataset

| n-shot Acc. | n=0 | n=1 | n=4 | $\tau$ |
|---|---|---|---|---|
| *Frozen* | 29.5 | 35.7 | 38.2 | ✗ |
| *Frozen* scratch | 0.0 | 0.0 | 0.0 | ✗ |
| *Frozen* finetuned | 24.0 | 28.2 | 29.2 | ✗ |
| *Frozen* train-blind | 26.2 | 33.5 | 33.3 | ✗ |
| *Frozen* VQA | 48.4 | – | – | ✓ |
| *Frozen* VQA-blind | 39.1 | – | – | ✓ |
| **Oscar [24]** | 73.8 | – | – | ✓ |

| n-shot Acc. | n=0 | n=1 | n=4 | $\tau$ |
|---|---|---|---|---|
| *Frozen* | 5.9 | 9.7 | 12.6 | ✗ |
| *Frozen* 400mLM | 4.0 | 5.9 | 6.6 | ✗ |
| *Frozen* finetuned | 4.2 | 4.1 | 4.6 | ✗ |
| *Frozen* train-blind | 3.3 | 7.2 | 0.0 | ✗ |
| *Frozen* VQA | 19.6 | – | – | ✗ |
| *Frozen* VQA-blind | 12.5 | – | – | ✗ |
| **MAVEx [44]** | 39.4 | – | – | ✓ |

Table 1: Transfer from Conceptual Captions to VQAv2. The $\tau$ column indicates whether a model uses training data from the VQAv2 training set. The row denoted *Frozen* train-blind is the blind baseline described in subsection 4.1. *Frozen* VQA is a baseline which mixes in VQAv2 training data.

Table 2: Transfer from Conceptual Captions to OKVQA. The $\tau$ column indicates if a model uses training data from the OKVQA training set. *Frozen* does not train on VQAv2 except in the baseline row, and it never trains on OKVQA.

that consists of around three million image-caption pairs [37]. We do early stopping on the validation set perplexity which usually reaches an optimum just after a single epoch with batch size 128. All experiments used the Adam optimizer with $\beta_1 = 0.9$ and $\beta_2 = 0.95$ and a constant learning rate of $3e\text{-}4$ unless otherwise noted. We operate on $224\times224$ images at both train and test-time. Images which are not square are first padded with zeroes to square and then resized to $224\times224$.

## 4.1 Rapid Task Adaptation

We first examine zero-shot and few-shot generalization from captioning to visual question-answering. This is a type of *rapid adaptation* from captioning behaviour to question-answering behaviour analogous to transfer from language modelling to open-domain question-answering in the text-only setting [34]. We evaluate on the VQAv2 [10] validation set.

**Zero-shot transfer from captioning to VQA** We first observe that a version of our model in which the ability to embed images into a prefix is trained solely with a captioning objective can transfer moderately well to visual question-answering in the zero-shot setting, with no specific training towards that goal. We simply have to provide the system with an image and a textual prompt of the form *Question: what colour is the dog sitting on the grass? Answer:*, then observe how it completes the prompt. The ability to adapt to input of this form is presumably transferred from the training data of the pretrained language model component of the system.

The strength of the pre-trained language model in the system is a double-edged sword. It powers the generalization abilities of *Frozen* but also enables the model to perform surprisingly well without considering the visual input at all. To guard against this possibility we also train blind baselines, in which the image presented to the visual encoder is blacked out, but the convnet weights are still trained (see Table 1). This amounts to prefix tuning [23]. We outperform this blind baseline which also inherits the few-shot learning abilities of the language model.

In these experiments we also include two additional and important baselines: *Frozen* finetuned in which the language model is instead finetuned starting from the pretrained weights and *Frozen* scratch, wherein the whole system is trained from scratch end-to-end, both using the same dataset as *Frozen*. These baselines preferred a smaller learning rate of $1e\text{-}5$. Results in Table 1 show that keeping the language model frozen generalizes substantially better to visual question-answering than finetuning. The model trained from scratch is not able to transfer at all from captioning to VQA; we interpret this to suggest that the tremendous generalization abilities of large language models are reliant upon large-scale training datasets in which the task of predicting the next token mimics the test setting (here question-answering) with non-negligible frequency.

**Improving performance with few-shot learning** More importantly, for the present work, we observe that the ability of the model to transfer knowledge from captioning and text-modelling to visual question-answering improves if the model is presented with several examples of VQA data sequentially. We repeat the previous experiments with up to four examples of image-question-answer

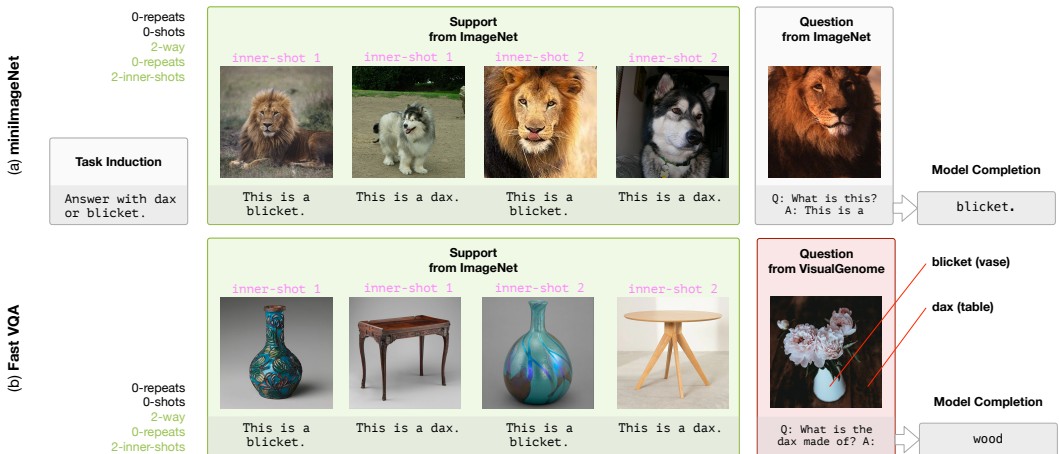

Figure 4: Examples of (a) the Open-Ended miniImageNet evaluation (b) the Fast VQA evaluation.

triples shown to the model as conditioning information in the continuous prefix sequence (using the interface in Figure 3).

These results are presented in Table 1. For contrast, we compare this performance to a condition in which we mix in some data from the VQAv2 training set with the captioning data. As we might expect, few-shot learning on four examples is outperformed by SGD on tens of thousands of examples, but few-shot performance clearly improves with more examples, and goes some way (38.2%) toward closing the gap from zero-shot performance (29.5%) to full SGD training performance (48.4%)

There are two important takeaways from the results presented in this section. First, they show that training a visual encoder through a pretrained and frozen language model results in a system capable of strong out-of-distribution (zero-shot) generalization. Second, they confirm that the ability to rapidly adapt to new tasks given appropriate cntext is inherited from the pretrained language model and transfers directly to multimodal tasks.

## 4.2 Encyclopedic Knowledge

Here we study the extent to which *Frozen* can leverage the encyclopedic knowledge in the language model towards visual tasks. The Conceptual Captions dataset is *hypernymed* (e.g. proper names are replaced with a general word like *person*). This enables us to rigorously study the transfer of factual knowledge because all knowledge of named entities comes from language model pretraining.

Consequently, when we show the model an image of an airplane and ask "who invented this?" (Figure 1), the visual encoder has determined that the image contains an airplane, and the language model has used this to retrieve the factual knowledge that airplanes were invented by the Wright brothers, a fact which is referenced in the C4 training set through (text-only) articles about airplanes. This is a fascinating chain of deduction. A detailed analysis of this behaviour with more examples is included in the Appendix (e.g. Figure 9, Figure 10, Figure 11).

We bolster this finding quantitatively by evaluating performance on OKVQA [27], a visual question-answering dataset designed to require outside knowledge in order to answer correctly. The pretrained language model's command of factual knowledge is of course dependent upon its scale, so we examine the performance of *Frozen* using pretrained language models of varying sizes: the base model with 7 billion parameters, and a much smaller 400 million parameter language model pretrained on the same dataset. Table 2 shows the results: task performance scales with model size. Again finetuning performs worse than leaving the model frozen in terms of generalization performance. We stress that *Frozen* is never trained on OKVQA.

## 4.3 Fast Word-to-Visual-Category Binding

In the multi-modal setting, fast-binding refers to a model's ability to associate a word with a visual category in a few shots and immediately use that word in an appropriate way.

**Open-Ended miniImageNet and Real-Name miniImageNet** To quantify the fast-binding capacity of of *Frozen*, we evaluate it on the miniImageNet meta-learning task [43]. Note that there are important differences with how we attempt miniImageNet and how it is approached in previous work. First, unlike standard meta-learning, we do not train *Frozen* on the (meta) task. Second, we evaluate *Frozen* in an open-ended fashion, where it must successfully generate a correct category name (and then the EOS token) in order to be credited with a correct answer. Finally, although we use the same image classes as the miniImageNet test set, they are at higher resolution ($224 \times 224$) and with integer class labels $[0, 1]$ replaced with nonsense words ('dax', 'blicket' etc). We make this adjustment because the nonsense words should have no (or less) intrinsic meaning to the language model than integers, whose relative order (for instance) should have been reflected in a massive text training corpus. We refer to this task as *Open-Ended miniImageNet*. To assess how much difficulty is added by binding visual categories to nonsense words versus simply adapting to an image recognition task *per se*, we also consider a version – Real-Name miniImagenet – in which visual categories in both the support set and the answer retain their original names. See Figure 4a for an illustration.

On both versions of this evaluation, we experiment by exposing the model to different numbers of inner-shots, repeats and task induction. On two-way Open-Ended miniImagenet, we observe that when *Frozen* is presented with a sequence of images and descriptions of new names for them, it is able to learn new names for the objects presented and then use these new names immediately with substantially above chance accuracy. Importantly, the ability of the model to use these new words improves with more examples of the corresponding category. Notably, this upward trend is more pronounced when this supporting information involves different exemplars from the visual category (inner-shots) rather than repetitions of a single exemplar (repeats). The fast-binding capacities of the model can thus be improved with richer and more varied visual support or prompting.

On two-way Real-Name miniImagenet, we observe a similar trend but with higher absolute performance. This underlines the difficulty in Open-Ended miniImagenet introduced by having to assign novel words to categories that may otherwise be already known to the model, and because the real names may carry visual information leveraged from the captioning data the model was trained on.

In Table 4, we show that the observed effects on Open-Ended miniImagenet do not transfer to the 5-way setting, where *Frozen* is not significantly above chance. This shows that learning to bind five new names to five visual categories in a single forward pass is beyond the current capabilities of *Frozen*. As before, however, we do observe an upward trend in the model's capacity to return the actual name for a visual category among the five possibilities as the number of inner-shots or repeats increases. Further work is required and we look forward to progress in this more challenging setting.

| Task Induction | ✗ | ✓ | ✓ | ✓ | ✓ | ✓ | ✓ |
|---|---|---|---|---|---|---|---|
| Inner Shots | 1 | 1 | 3 | 5 | 1 | 1 | 1 |
| Repeats | 0 | 0 | 0 | 0 | 1 | 3 | 5 |
| *Frozen* | 29.0 | 54.1 | 55.2 | 57.6 | 51.8 | 57.7 | 58.6 |
| *Frozen* (**Real-Name**) | 1.7 | 49.2 | 67.0 | 68.4 | 63.8 | 65.2 | 64.0 |
| *Frozen* test-blind | – | 48.5 | 46.7 | 45.3 | – | – | – |
| *Frozen* test-blind (**Real-Name**) | – | 1.0 | 12.6 | 33.0 | – | – | – |
| **ANIL Baseline [32]** | – | 73.9 | 81.7 | 84.2 | – | – | – |

Table 3: Performance of *Frozen* and baselines on Open-Ended miniImageNet 2-Way Tasks. Randomly picking between the two class labels (then emitting the EOS token) would yield 50% accuracy. As the model has to generate the answer, and is not counted correct if it paraphrases, this is not the best blind baseline, which is why we include open-ended blind baselines that also generate.

**Fast-VQA and Guided-VQA** As transformers are trained to model text, their attention weights learn to associate – or 'bind'– pairs of words across sentences. The experiments with miniImageNet show that this capacity can transfer directly to binding visual categories to their names, enabling the system to generate the name on demand. This raises the question of whether *Frozen* can integrate a newly-acquired visual category (and its names) more fully into the model's language system, so that it can, for instance, describe or answer questions about that category.

To test this capacity, we constructed a new task – *Fast-VQA* – out of two well-known datasets, ImageNet [35] and Visual Genome [17]. For each question, the model is presented with nonsense

| Task Induction | ✗ | ✓ | ✓ | ✓ | ✓ | ✓ | ✓ |
|---|---|---|---|---|---|---|---|
| Inner Shots | 1 | 1 | 3 | 5 | 1 | 1 | 1 |
| Repeats | 0 | 0 | 0 | 0 | 1 | 3 | 5 |
| *Frozen* | 18.0 | 21.3 | 22.4 | 22.1 | 21.5 | 21.1 | 20.9 |
| *Frozen* (Real-Name) | 0.9 | 12.4 | 34.0 | 31.0 | 32.0 | 33.2 | 33.8 |
| *Frozen* test-blind | – | 18.6 | 19.9 | 19.8 | – | – | – |
| *Frozen* test-blind (Real-Name) | – | 4.6 | 22.6 | 20.8 | – | – | – |
| ANIL Baseline [32] | – | 45.5 | 57.7 | 62.6 | – | – | – |

Table 4: Performance of *Frozen* and baselines on Open-Ended miniImageNet 5-Way Tasks. Randomly picking between the five class labels (then emitting the EOS token) would yield 20% accuracy.

words ('dax' and 'blicket') and $n$ images of the referents of those words (e.g. of a 'cat' or a 'dog') taken from ImageNet. It is then asked a question containing at least one of those two words, about a further image (taken from Visual Genome) in which *both* of the referents appear (see Figure 4b). As with miniImagenet, the words 'dax' and 'blicket' (and how they refer) should be new to *Frozen*, but the corresponding visual categories may be known from the Conceptual Captions training data, albeit by different names.

To quantify how much harder the introduction of new words for known categories makes this task, we also created a variant (*Guided-VQA*) in which the original category names ('cat' or 'dog') are used instead of 'dax' and 'blicket'. Guided-VQA is a special case of Fast-VQA involving questions from Visual Genome, where the model is reminded what the important entities in the question look like prior to answering the question by labeling sample images with real category names. *Guided-VQA* does not require the same ability to bind categories to new words, but it does measure how well a model can exploit task-relevant multimodal guidance when attempting a new task in an otherwise zero-shot manner.

Fast-VQA and Guided-VQA are very challenging tasks because they are attempted without task-specific training, and because the underlying questions come from Visual Genome (VQAv2 images do not come with the necessary meta-data to construct the task). Visual Genome questions are particularly challenging because only a single answer exists for each question. When scoring models, for simplicity we credit only an exact match with the output generated by the model, modulo the same post-processing applied for VQAv2. Because of the inherent difficulty of the task, we use strong baselines that can still utilize the large language model to verify strength of observed effects.

| | Fast-VQA | | | | Guided-VQA | | | |
|---|---|---|---|---|---|---|---|---|
| Inner Shots | 0 | 1 | 3 | 5 | 0 | 1 | 3 | 5 |
| *Frozen* | 1.6 | 2.8 | 7.0 | 7.9 | 3.7 | 7.8 | 10.1 | 10.5 |
| *Frozen* train-blind | 0.7 | 0.3 | 1.3 | 0.4 | 1.9 | 2.3 | 3.7 | 3.7 |

Table 5: Performance of *Frozen* versus an equivalent blind model on Fast and Guided-VQA.

As shown in Table 5, the fact that the model improves with more shots in both Fast-VQA and Guided-VQA confirms that *Frozen* has some capacity to integrate novel words into its general capacity to process and generate natural language in a multimodal context. It is notable that a prefix-tuned model with no access to images improves moderately at Guided-VQA as more categories are presented, showing that additional linguistic cues (just being reminded of the words involved and the linguistic form of the task) goes some way to preparing for the upcoming question. As exemplified in Figure 4, inspection of the model output confirms that in many cases it is indeed the multimodal (and not just linguistic) support that enables *Frozen* to improve performance as the number of shots increases. We observed that there are diminishing returns in performance gain as the number of shots increase. One possible explanation is that the shift from the training distribution of contexts with single images to multiple images causes inaccuracies in the model.

The Open-Ended miniImagenet, Real-Name miniImagenet, Fast-VQA and Guided-VQA evaluation sets are available to download at `https://fh295.github.io/frozen.html`.

# 5 Discussion

## 5.1 Limitations

We believe this work is an important proof-of-concept for a desired, much more powerful system capable of open-ended multimodal few-shot learning. *Frozen* achieves the necessary capacities *to some degree*, but a key limitation is that it achieves far from *state-of-the-art* performance on the specific tasks that it learns in a few shots, compared to systems that use the full training set for those tasks. As such, the main contribution of this work should be seen as a starting point or baseline for this exciting area of research of multimodal few-shot learning.

Further improvement can make the impressive zero-shot and few-shot generalization we observed more robust as reflected by higher accuracy and fewer seeds required to demonstrate our most compelling samples. Finally, there are many technical questions that were not explored in this proof-of-concept study, such as whether performance could be improved with more elaborate architectures for mixing vision and language. We leave the exploration of these possibilities to future investigations. The Open-Ended miniImageNet, Real-Name miniImagenet, Fast-VQA and Guided-VQA benchmarks that we provide with this manuscript should facilitate the evaluation and analysis of future systems of this type.

## 5.2 Societal Impact

With the emergence of this new class of general purpose vision-language models, new capabilities of massive surveillance can be feasible. Both surveillance footage and publicly shared images can be analyzed for arbitrary questions without requiring any new labeled data or training of the system. As a mitigation for individuals, personal assistant software with similar capabilities can analyze publicly available documents about themselves to identify unintended exposures, even when novel concerns emerge either due to societal change or change of personal preferences.

Generative models of text that can incorporate visual information can elevate the misuse of language-model generated content by making them even more convincing. Moreover, at this point we do not have sufficient tools to identify bias and toxicity issues of general purpose vision-guided language models. We invite the community to think about effective methods and benchmarks on this front.

More positively, systems like Frozen could be applied to assist visually impaired users of technology. Frozen's ability to adapt to different styles of caption or question could enable a more personalised user experience in these cases.

There are environmental costs associated with training the large networks in systems like Frozen. On the other hand, a system that can be trained once and then flexibly adapted to different settings could have a lower energy footprint overall than one that requires re-training for different applications.

## 5.3 Conclusion

We have presented a method for transforming large language models into multimodal few-shot learning systems by extending the soft-prompting philosophy of *prefix tuning* [23] to ordered sets of images and text while preserving text prompting abilities of the language model. Our experiments confirm that the resulting system, *Frozen*, is capable both of open-ended interpretation of images and genuinely multimodal few-shot learning even though the system is only trained to do captioning. One corollary of these results is that the knowledge required to quickly bind together or associate different words in language is also pertinent to rapidly binding language to visual elements across an ordered set of inputs. This finding extends the conclusion of [26] – that knowledge in transformer language models can transfer to non-linguistic tasks – to the specific case of knowledge about few-shot learning.

**Acknowledgements** We wish to thank Sebastian Borgeaud and Jack Rae for preparing the pre-training text dataset and pretraining a selection of transformer language models, as well as Trevor Cai for help with experiments and infrastructure. We also wish to thank Pauline Luc, Jeff Donahue, Malcolm Reynolds, Andy Brock, Karen Simonyan, Jean-Baptiste Alayrac, Antoine Miech, Charlie Nash, Aaron van den Oord, Marc Deisenroth, Aida Nematzadeh, Roman Ring, Francis Song, Eliza Rutherford, Kirsty Anderson, Esme Sutherland, Alexander Novikov, Daan Wierstra, and Nando de Freitas for insightful discussions during the course of the project.

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
