# A  Appendix

## A.1  Compute Usage

The seven billion parameter language model we used as part of *Frozen* used model parallelism with the strategy from [39] to partition one instance of the model over four accelerators. Each instance had a batch size of 8. To reach a batch size of 128 in this configuration, we additionally employed data parallelism with 16 synchronous replicas. The whole system was trained on a 4x8 TPUv3 [15] topology for about 12 hours, which is when validation set performance for Conceptual Captions led us to do early stopping.

## A.2  Frozen Architecture Details

The pretrained transformer language model we used has a GPT-like architecture [30]. It consists of a series of identical residual layers, each comprised of a self-attention operation followed by a positionwise MLP. The only deviation from the architecture described as GPT-2 is the use of relative position encodings [38]. Our seven billion parameter configuration used 32 layers, with each hidden layer having a channel dimensionality of 4096 hidden units. The attention operations use 32 heads each with key/value size dimensionality of 128, and the hidden layer of each MLP had 16384 hidden units. The 400 million parameter configuration used 12 layers, 12 heads, hidden dimensionality of 1536, and 6144 units in the MLP hidden layers.

## A.3  Few-Shot Learning Definitions

As *Frozen* can be conditioned on a sequence of interleaved images and text, it is capable not only of performing on a variety of multimodal tasks, but also, the same task can be induced in multiple ways to help *Frozen* to learn and perform better. In order to make it easier to distinguish among these different ways of 'inducing' a task to the model, we have formalized the terminology used in our settings, which is described in section 3.4 of the main text. In Figure 5 and Figure 6 below we provide more visual examples of this terminology.

## A.4  Tasks to Evaluate Fast-Binding Capacity

### A.4.1  Open-Ended MiniImageNet

To construct the Open-Ended MiniImagenet evaluation we begin with the same subset $S$ of ImageNet classes applied in prior on meta-learning with MiniImagenet (See the appendix of [33]). All images are taken from the validation set of ImageNet.

To generate a 2-way question with $n$ inner-shots, the following process is followed:

1. Sample two classes $c_1, c_2$ from $S$
2. Sample $n$ images $v_1^{c_1} \ldots v_{n+1}^{c_1}$ from $c_1$ and $n$ images $v_1^{c_2} \ldots v_n^{c_2}$ from $c_2$
3. Interleave into a sequence of $2n$ support images $[v_1^{c_1}, v_1^{c_2} \ldots v_n^{c_1}, v_n^{c_2}]$
4. Assign the nonsense words (*dax, blicket*) to $c_1, c_2$ at random, and interleave support captions *"this is a dax"* or *"this is a blicket"* accordingly
5. Select one of $c_1, c_2$ at random, $c_q$, and sample a further question image $v^{c_q}$
6. Assign the truncated caption *"this is a"* to $v_q$ and the appropriate nonsense word as the correct answer.

Note that this process ensures that the image class and nonsense word assigned to the correct answer occur in either first or second place in the support, and the correct answer may be *dax* or *blicket* with equal probability.

To generate a 5-way question, the above process is generalized. In 1. five distinct classes are sampled from $S$. The set of nonsense words applied in step 4. and 6 is: [*dax, blicket, slation, perpo, shously*]. The final three words were taken from a nonsense-word generator[1] and selected because, like *dax* and *blicket* and for consistency, they decompose into two tokens in our model's subword vocabulary.

---

[1] https://www.soybomb.com/tricks/words/

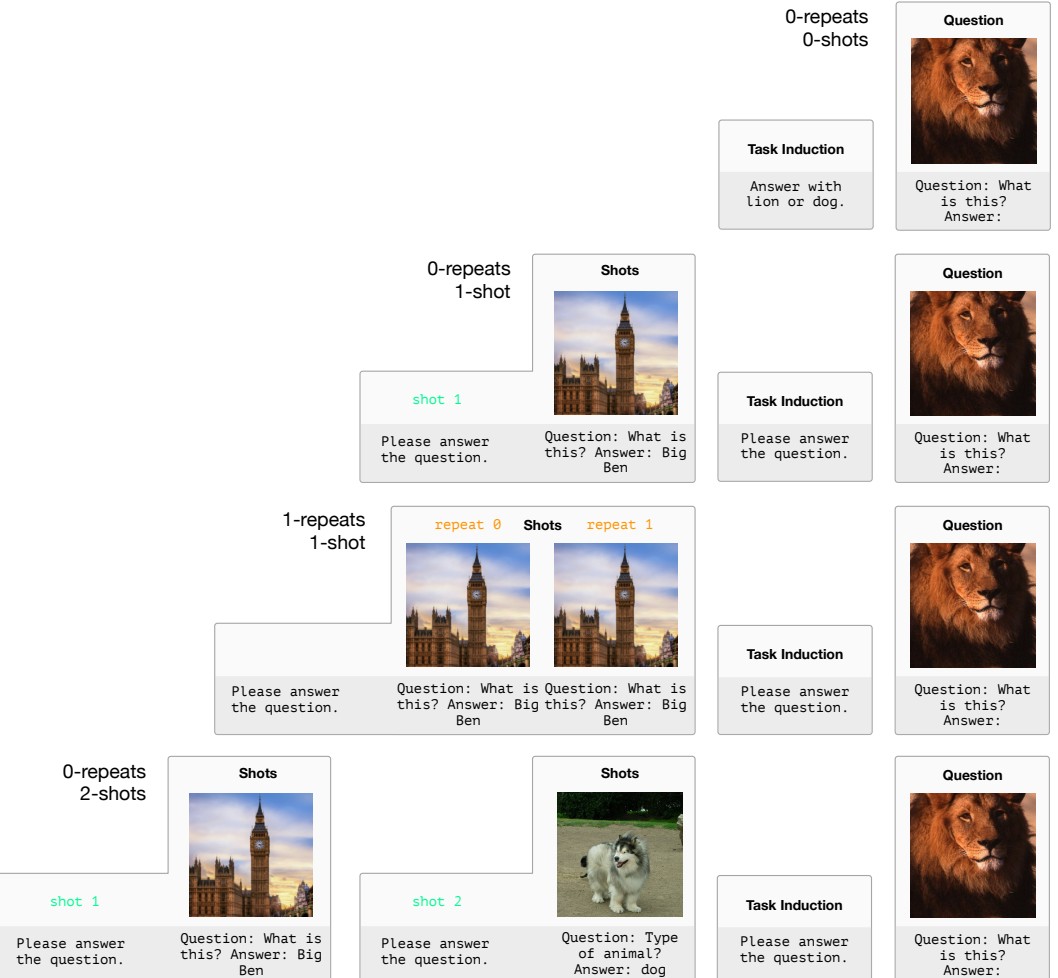

Figure 5: Examples of few-shot learning vocabulary.

All images are stored at $224 \times 224$ resolution.

### A.4.2 Real-Name miniImageNet

To generate Real-Name miniImagenet, the same process is followed, except that in steps 4. and 6., instead of using nonsense words to caption the support images (e.g. *"this is a dax"*), the (first) class name from the ImageNet dataset is used (e.g. *"this is a fruit bat"*).

### A.4.3 Fast-VQA

Unlike Open-Ended miniImageNet, Fast-VQA uses images from all 1,000 classes in the ImageNet dataset. For the evaluations in this paper, we again only take images from the validation set. Denote by $W$ the set of all 1,000 class (first) names, and for each $w_i \in W$, the corresponding set of images $c_i$.

The Visual Genome (VG) dataset contains meta-data, questions and answers, such that we can consider data in the form $(Im, q, a, Ob)$, where $Im$ is the image, $q$ is the corresponding question, $a$ is the answer and $Ob$ is a list of names for all objects annotated in $Im$. We first filtered the dataset into a subset $VG*$ such that every question $q_k$ contained at least one word $w_i \in W$ *and such that* the corresponding object list $Ob_k$ also contained $q_k$ and at least one other word $w_j \in W : w_j! = w_i$. Thus, we can consider the elements of $VG*$ to be of the form $(Im, q, a, Ob, w_i, w_j)$

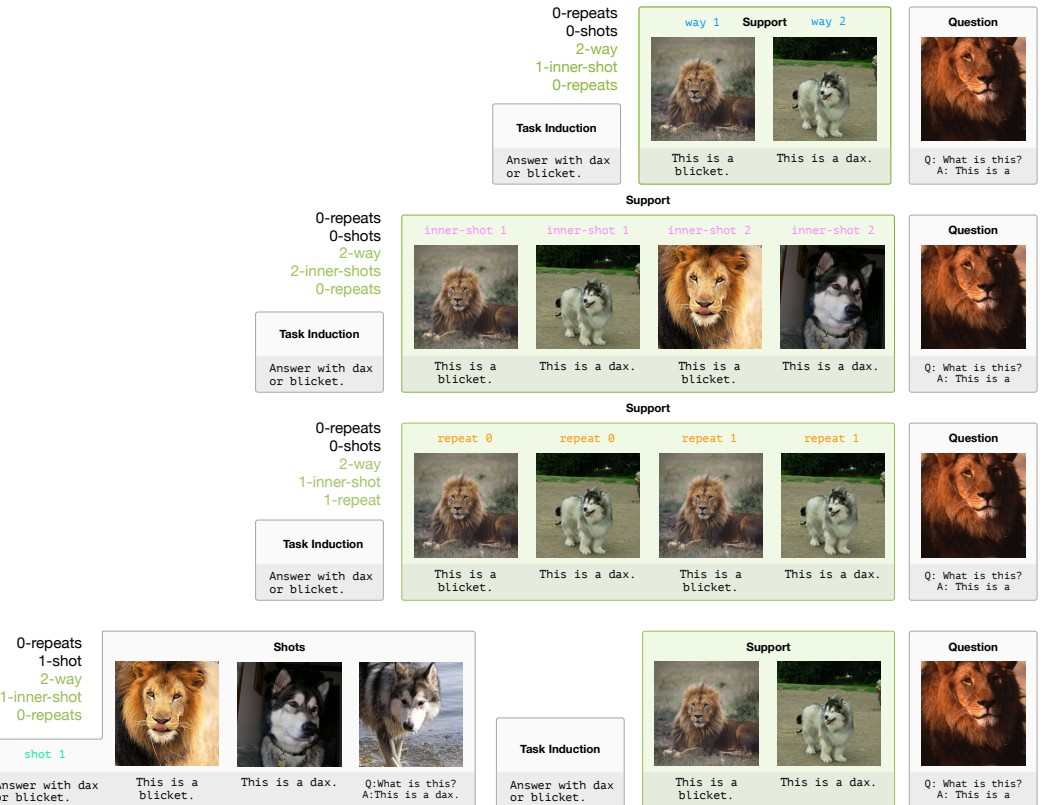

Figure 6: Examples of few-shot learning vocabulary for fast-binding.

To generate a 2-way, $n$-shot Fast-VQA question out of an element $(Im, q, a, Ob, w_i, w_j)$, we then did the following:

1. Sample $n$ images $v_1^{c_i} \ldots v_{n+1}^{c_i}$ from $c_1$ and $n$ images $v_1^{c_j} \ldots v_n^{c_j}$ from $c_2$

2. Depending on coin toss, form either the support $[v_1^{c_i}, v_1^{c_j} \ldots v_n^{c_i}, v_n^{c_j}]$ or the support $[v_1^{c_j}, v_1^{c_i} \ldots v_n^{c_j}, v_n^{c_i}]$

3. Assign the nonsense words (*dax*, *blicket*) to $w_i, w_j$ at random, and interleave support captions *"this is a dax"* or *"this is a blicket"* accordingly

4. Transform $q$ and $a$ into modified questions and answers $q*$ and $a*$ by replacing all instances of $w_i$ and any instances of $w_j$ with the corresponding strings *dax* or *blicket*

5. Append the (VG) question $(Im, q*, a*)$ to the (ImageNet) support from 2. to create the Fast-VQA sample.

In this work, we only consider 2-way Fast-VQA.

### A.4.4 Guided-VQA

To generate Guided-VQA, the same process is followed, except that in step 3. the (first) class name from ImageNet is used to caption the support images (*"this is a cat"*, *"this is a wolf"*), and no string replacement is undertaken in 4.

The Open-Ended miniImageNet, Real-Name miniImageneNet, Fast-VQA and Guided-VQA evaluations are available at `https://fh295.github.io/frozen.html`.

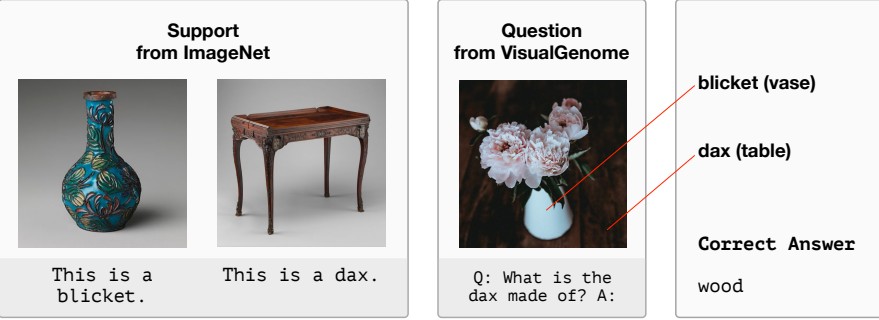

Figure 7: Example of a Fast-VQA task.

## A.5 Encyclopedic Knowledge

Here we add more detail to the claim in subsection 4.2 that the model seems to be performing a sort of multi-hop deduction in the "Wright Brothers" example from Figure 1.

First, there has been a substantial amount of recent work studying a language model's ability to draw upon factual knowledge, examining the ability of language models to answer factual questions either zero-shot [28, 4] or after open-domain QA finetuning [34, 11, 21]. Buoyed by these findings, we here demonstrate rigorously the impressive extent to which *Frozen* seems to be commanding this factual knowledge and drawing upon it when prompted by an image (here an image of an airplane). We now break down why it is interesting that the model correctly determines that the Wright Brothers invented the object in the image (an airplane), by studying how the model responds to different prompts concerning this same test image in Figure 9.

Recall that Conceptual Captions is *hypernymed* so none of the language targets used to train *Frozen* contain named entities like "The Wright Brothers". Instead, our training signal teaches the model to emit text that would roughly describe an image. The impressive finding is that this scalable, weakly supervised objective *generalizes* to general information retrieval about an image.

The top pane in Figure 9 shows an example of what the text in the captioning distribution looks like, captioning the image as "an airplane flying over a blue sky – stock photo #". Now, as established in subsection 4.1 we enjoy some amount of zero-shot transfer from captioning to visual question-answering. This is demonstrated in the second and third rows of Figure 9. But, adhering to the distribution of caption text, the model does not give a named entity when asked who invented the airplane. Instead it completes the prompt vaguely by saying "This was invented by *an aerospace engineer and is made by the brand he worked for*".

But we know for certain that the language model has learned plenty of facts about named entities during pre-training and in particular we determined via the C4 dataset search tool [9] that there are multiple articles concerning the Wright Brothers. It's just that matching the distribution of Conceptual Captions text has taught the model to not emit named entities when prompted with an image. But the model can *recover* the ability to refer to named entities given an image with few-shot learning (bottom row of Figure 9). We show the model two examples of saying who invented an object depicted in an image by giving a named entity (Zacharias Janssen invented the microscope and Henry Ford invented the model T, an early automobile). With this prompt, *Frozen* reliably retrieves the correct factual knowledge, having determined in the vision encoder that the image depicts an airplane, and having been demonstrated in-context that the desired output is the name of a person.

This outcome is robust, in the sense that we observed it in multiple versions of *Frozen* during development, and in multiple examples, but drawing samples is not always successful and can require 3-4 tries to get past well-known language model failure modes of either repeating text from its prompt or emitting completely unrelated text. That's why we describe some samples as "curated".

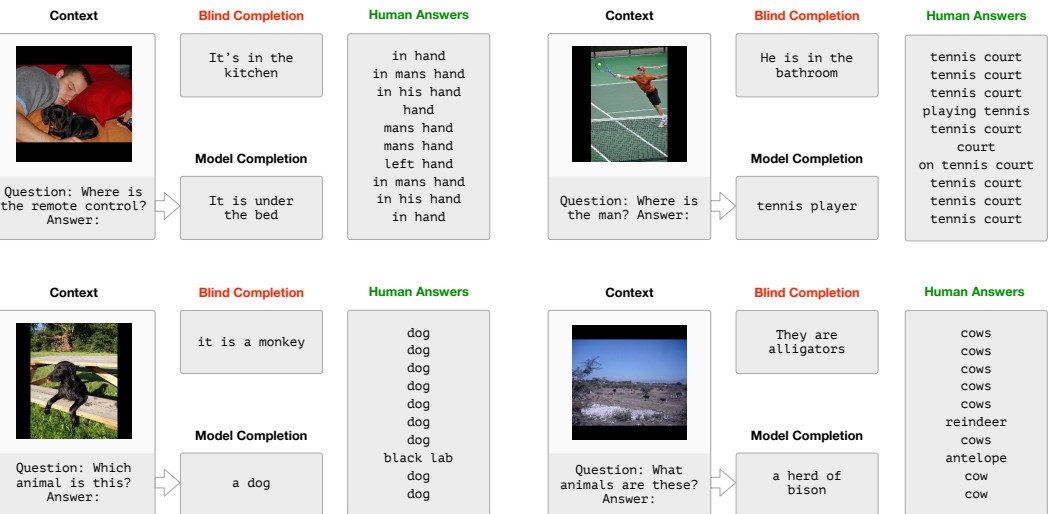

Figure 8: VQA qualitative. This is a greedy sample of our model's prediction on a VQAv2 validation set example. See accuracy numbers in Table 1 for overall robustness.

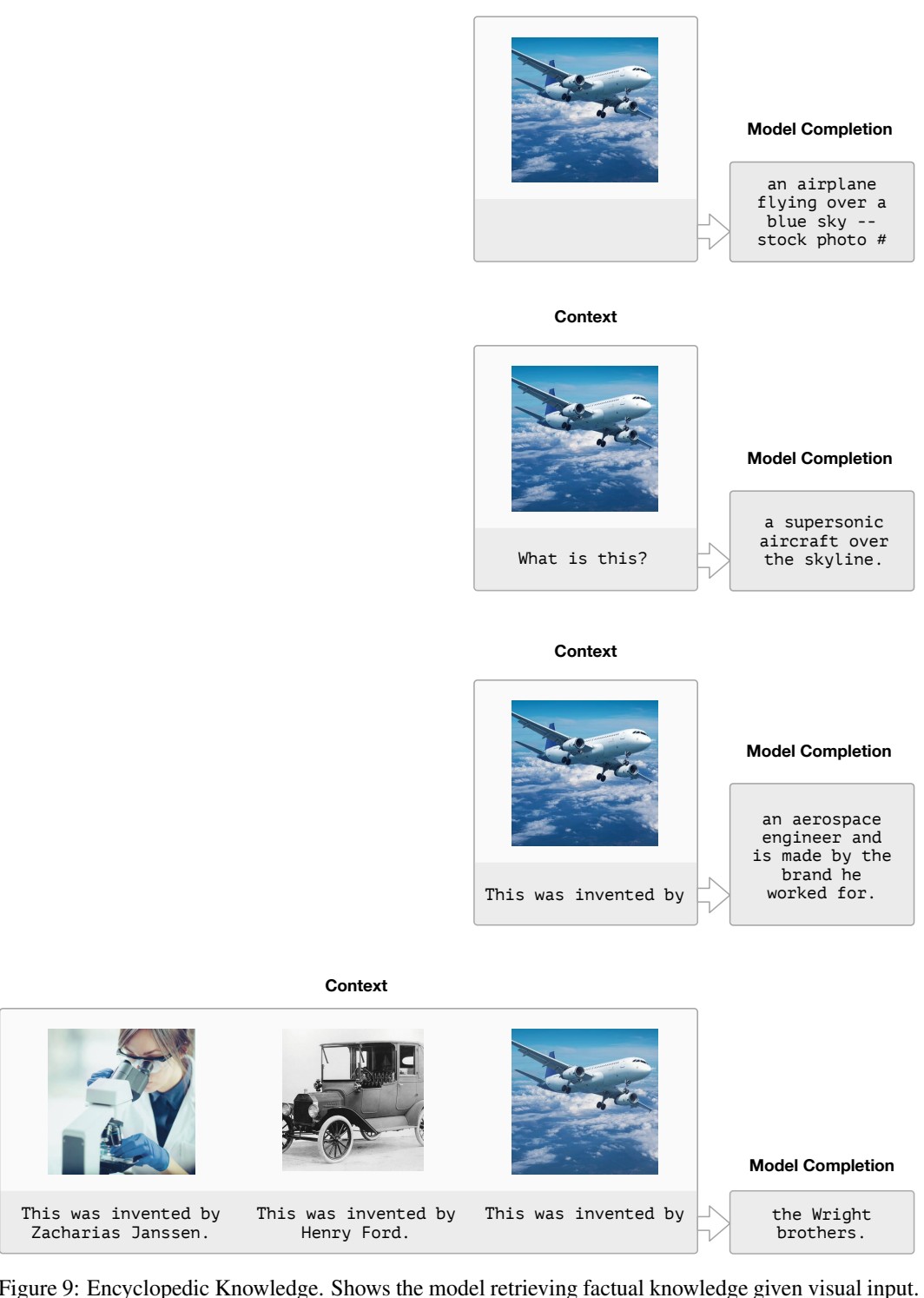

Figure 9: Encyclopedic Knowledge. Shows the model retrieving factual knowledge given visual input. Required cherry-picking from around 5 seeds to get past common language model failure modes like simply repeating text from the prompt or emitting text that does not pertain to the test image.

**Context**

This is a 🐴.   This is a 🐱.   This is a

**Model Completion**

🐶. <EOS>

Figure 10: Emojis. This sample reliably produced good output within a few attempts but did not work for every seed.

**Context**

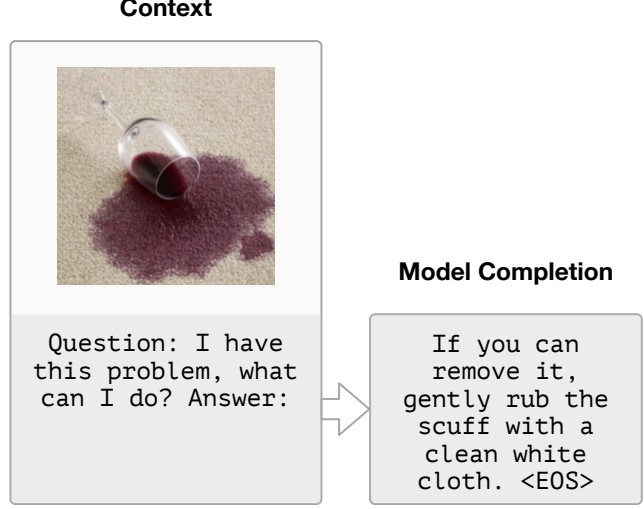

Question: I have this problem, what can I do? Answer:

**Model Completion**

If you can remove it, gently rub the scuff with a clean white cloth. <EOS>

Figure 11: Encyclopedic Knowledge. Demonstrates knowledge from language pre-training being commanded given visual input. Required a few seeds to get a good answer which clearly paid attention to the image.