# OpenReview forum: "Multimodal Few-Shot Learning with Frozen Language Models"
_NeurIPS.cc/2021/Conference — NeurIPS 2021 Poster_

### Official Review · Reviewer_qf5a · 2021-07-09

**Rating:** 6
**Confidence:** 4

**Summary:**

The paper proposes a new GPT-2 style image-language model, Frozen, which is trained in two stages:
1. common GPT-2 pre-training on huge text-alone corpora
2. training of a Resnet image encoder while freezing the GPT-2 and text embedding weights, on the task of image captioning (i.e. language modeling conditional on two image encoding vectors with the same dimension as token embeddings).

Since GPT-2 exhibits few-shot abilities in language-alone tasks, the idea is Frozen can also benefit from the GPT-2 language pertaining and exhibit *multimodal* few-shot abilities by the minimal design of adding a visual encoder. Note that the freezing does not make training more efficient, since gradients have to pass through all Transformer layers to the visual encoder.

The Frozen model is applied to zero/few-shot learning of two downstream tasks: VQA and few-shot image classification.

**Ethical Concerns:**

No.

**Limitations And Societal Impact:**

Unfortunately, the results seem too low to be claimed surprising. With better experiments I'd definitely raise the score.

- For VQA (Table 1), the gap between Frozen and Frozen (train-blind) seem very small, which suggests maybe the majority of correctly answered questions are thanks to language-alone learning? I'd like to see some non-trivial examples.

- For few-shot classification (Table 3, 4), the accuracies are merely 50-something and 20-something percents for two and five way classification, while random baseline would achieve 50% and 20%. This makes me hard to believe Frozen achieves "fast concept binding". Besides, I am not sure if the Real-Name result (66%/34%) can prove the model achieves "slow concept binding" either, given the baseline's 84%/62% performance.

- Given the poor lag behind visual-language baselines, is it possible to make a fairer comparison by controlling the training data to Frozen and baselines? For example, make sure Frozen and Oscar receives the same amount of training data (text corpora for language-alone pretraining, paired image-captions for visual encoder training), and then study the zero/few-shot test case? Showing the baselines performing a lot better with more training data isn't very helpful to determine how Frozen behaves.

- Conceptually, why would language learning first enable visual learning later? Why not the other direction (visual learning first, language learning later), given that's more like how humans learn? I'm also curious about such a Frozen variant: mix the two training stages (language modeling, language modeling conditional on visual features) together and learn GPT-2 and visual encoder together. If such a variant is worse than Frozen, it better justifies the two-stage approach with freezing.

- Is the visual encoder pre-trained anyway? It is not clearly stated in the main paper. If not, it might be interesting to evaluate the visual encoder alone, testing how much better at vision tasks it has been than a random encoder, thanks to the feedbacks from the language model. I'm also curious what if a pre-trained ResNet is used (frozen or fine-tuned) what will happen. If so, what type of pre-training is used, based on what data?

**Main Review:**

**Significance**: The work is at the intersection of two important topics: multimodal (vision-language) learning and few-shot learning. Given the surprising zero/few-shot abilities of recent large language models (e.g. GPT-2/3, T5), it is natural to investigate how such abilities can be instantiated in a multimodal setup, which requires the learner not only for rapid task adaption and knowledge transfer, but also fast binding between visual features and language concepts. Such a study should be very impactful and pertaining both to cognitive science and AI, given how humans possess such strong abilities but not machines yet.

The work also relates to an important debate in NLP: can language models learn meanings from form (text corpora) alone? By leveraging a frozen language model to "heavy-lift" a visual encoder head, it probes to what extent can language learning enable learning and binding to visual concepts.

**Originality**: The work is interesting and original.

**Clarity**: The paper is well written and the main idea is simple and easy to follow. The figures are illustrative, though I was *slightly* confused by the left part of Figure 4 and what "0-repeats, 0-shots, ..., 0-repeats, 2-inner-shots" mean.

**Quality**: It is interesting how few-shot enables some boosts in results compared to zero-shot. However, my main concern is the poor experimental results in general. Please see below.


**Time Spent Reviewing:**

5

---

> ### Author Response · Authors · 2021-08-11
> **Response to Reviewer qf5a**
>
> Thank you very much for your review. We recognise your major concern, which we have addressed in the sibling, top-level thread for all reviewers to read. Please let us know if you have further questions on this point.
>
> Please also see below for answers to your specific questions.
>
> > Given the poor lag behind visual-language baselines, is it possible to make a fairer comparison by controlling the training data to Frozen and baselines? For example, make sure Frozen and Oscar receive the same amount of training data (text corpora for language-alone pretraining, paired image-captions for visual encoder training), and then study the zero/few-shot test case? Showing the baselines performing a lot better with more training data isn't very helpful to determine how Frozen behaves.
>
> We agree that this type of comparison could make Frozen look better. But it can be difficult to do such a comparison in a completely fair way as Frozen and Oscar (for example) are methods of training as well as architectures. The language-model (pre)-training data is an inherent part of Frozen, while we would have to adapt the Oscar algorithm in some creative way in order to learn from such data. Our preferred approach instead was to be completely transparent about the task-specific performance gap between Frozen and baselines whilst being clear that Frozen is playing a completely different game.
>
> > Conceptually, why would language learning first enable visual learning later? Why not the other direction (visual learning first, language learning later), given that's more like how humans learn? I'm also curious about such a Frozen variant: mix the two training stages (language modeling, language modeling conditional on visual features) together and learn GPT-2 and visual encoder together. If such a variant is worse than Frozen, it better justifies the two-stage approach with freezing.
>
> Many papers, including the first works on image captioning with neural networks, employ pre-trained vision encoders and train a language model from scratch. In this work, we were instead interested in what would happen if we adapted an untrained vision network to the needs of a pre-trained language model. The result was the surprising multimodal few-shot learning that we report and quantify.
>
>
> In this work, our goal has not been to imitate human learning. Instead, we wished to investigate whether large language models that typically require enormous resources to train (and can only be trained by the largest of research labs) can be used off-the-shelf to also learn vision. By doing so our hope is that the effort put into training large language models can be exploited in an efficient manner to obtain the multimodal models that we ultimately desire. As suggested in the next question, investigating pre-trained vision encoders can be a useful followup study.
>
> > Is the visual encoder pre-trained anyway? It is not clearly stated in the main paper. If not, it might be interesting to evaluate the visual encoder alone, testing how much better at vision tasks it has been than a random encoder, thanks to the feedback from the language model. I'm also curious what if a pre-trained ResNet is used (frozen or fine-tuned) what will happen. If so, what type of pre-training is used, based on what data?
>
> Our visual encoder is trained from scratch, but the suggestion to explore using also pre-trained vision encoders is a good one. Doing so would raise further questions that would have to be considered carefully, such as what is the correct pre-training objective for the visual encoder, which is why we felt it out of scope for the current manuscript. There is certainly a lot more to explore in this direction however, and we hope that we and others will do so in future work.

---

> > ### Comment · Reviewer_qf5a · 2021-08-17
> > **Re**
> >
> > Thanks for the rebuttal, which does address some of my concerns!
> >
> > - I agree Frozen and other models (OSCAR, ANIL) are playing different games, and it might be hard to control the data used in a fair way. I just expect the performance gap to be smaller.
> >
> > - Following the previous one. "we believe the fact the system is capable of any few shot learning at all (by outperforming strong baselines in a few-shot setting on VQA-style tasks and significantly above-chance on meta-learning benchmarks like miniImagenet) is a notable result". Given the success of GPT series and prompting, as a NLP guy I don't really think multi-modal few-shot ability would be very surprising. Plus, for VQA what is the "strong" baseline? For classification how to define "significantly" above-chance, given a sizable model is used?
> >
> > - The language-first motivation is reasonable, and thanks for agreeing with the visual encoder suggestion. These are two optional things that will make the current analysis stronger to me, given lack of effective model comparisons in the current version.

---

> > > ### Author Response · Authors · 2021-08-18
> > > **Response to Reviewer qf5a**
> > >
> > > Thank you for the followup and comments! We're pleased that the rebuttal addresses some concerns, and hope to address the remainder now.
> > >
> > > By 'outperforming strong baselines', we were referring to a large language model trained on the distribution of image captions (Frozen train-blind in Table 1). We considered this baseline 'strong' because it was non-trivial to surpass it - for instance Frozen with a relatively small LM performs at the level of this 'blind' baseline with a larger LM. This tells us something about the power of large LMs to adapt to the language distribution of new tasks, and something about how (un)important visual information is for a lot of VQA questions -- this is indeed highlighted in the original vqa paper where they measure the performance by training only on the language part of the task. It also confirms that the sort of multimodal few-shot learning that we observe in Frozen is a phenomenon that emerges at scale. We agree that 'strong' wasn't quite the right word to use for this baseline though - apologies if this was misleading.
> > >
> > > We used the term 'significant' because in conducting this research we trained several models and each time observed performance on the fast-binding tasks that was above random guessing. For instance, each of the three Frozen models we trained (with different hyperparameters) and evaluated on Open-Ended Mini-Imagenet scored around 10 percentage points above 50% random guessing. Just to confirm that this is indeed a significant effect, we quickly compared the mean of this sample of three models with the 50% level using a one-sample t-test.  With one shot of the task, the models did not reach a threshold of significance (M = 52.03, p=0.10) but for three (M = 58.77, p = 0.005) and five shots (M = 61.63, p = 0.014) they did. We couldn't put such analyses in the paper because we didn't train multiple models in each of the baseline conditions (to save compute). We also like to emphasize that all results for Frozen in the paper come from a single trained set of weights.
> > >
> > > Finally, we acknowledge your sense that the multimodal few-shot learning isn't too surprising given what these language models can do already. We will try to explain our perspective on this, and will emphasize these thoughts more clearly in the discussion of the paper. First, the only visual data that Frozen is trained on is single images and their corresponding captions. We were not expecting much when we first presented Frozen with sequences of multiple images and interleaved text - but when we probed it with hand-picked examples we observed that it produced reasonable continuations that in some cases seemed to integrate information from each of the images and passages. We found this to be a notable insight worth sharing, and we hope it will pave the way for future research. Note that this, in a pure NLP setting, would be equivalent to training a language model ONLY on single sentences, and expect it to perform well on full paragraphs or multiple paragraphs. In this sense, even with what large LMs have achieved, this is still non-trivial and surprising. Second, while the strength/robustness of the 'few-shot-learning' is indeed below text-only settings with conventional LMs, we don't think this is a fair barometer for what to expect here. Many text-based 'tasks' do presumably appear in sequence across the internet; so a LM gets some benefit (during training) from improving at them rapidly. There is nothing in the image captioning objective that prepares Frozen in the same way for multimodal few-shot learning - for this to work, its visual input must interface seamlessly with the text-based few-shot learning in the LM. We think, at least for those in the vision community, these results will be unexpected; and we hope we can agree that they are at least novel, interesting and intriguing.

---

> > > > ### Comment · Reviewer_qf5a · 2021-08-23
> > > > **Re**
> > > >
> > > > Thanks for the detailed followup! I think the paper will benefit from clarifications and discussions in our interactions, and I am happy to raise my score now.

---

> > > > > ### Author Response · Authors · 2021-09-02
> > > > > **Thank you.**
> > > > >
> > > > > Thank you very much for reconsidering your score. We agree that these clarifications will make the paper better.

---

### Official Review · Reviewer_DtyQ · 2021-07-15

**Rating:** 7
**Confidence:** 3

**Summary:**

The paper proposes to utilize the zero-shot/few-shot generalization ability of frozen language models in a multi-modal setting, by projecting visual input features into the text input feature space, through moderate pre-training on Conceptual Captions (the LM is frozen during pre-training).

The paper designs several interesting experiment settings to show the model’s ability of task adaptation, utilizing pre-trained encyclopedia knowledge, and novel concept binding.

My main concern is that it is hard to judge whether the results in these experiments are significant or somewhat trivial. But given the novelty of the setting and the promise, I would lean towards acceptance but I look forward to the author’s response.

**Ethics Review Area:**

["I don’t know"]

**Limitations And Societal Impact:**

Nothing I could think of.

**Main Review:**

Pro:

The picture the paper tries to paint is appealing and makes sense. By projecting the visual features into the same features space as the input embedding space of a frozen LM, it is reasonable to expect that the Frozen LM can make use of this visual input information and we could tap into the great zero-shot/few-shot generalization ability of the Frozen LM.

The paper presents three interesting experiment settings that truly explore the potential usage of the model, i.e. to quickly adapt to new tasks and new concepts.



Cons:
1. The paper somehow falls short as it does not deliver the same level of zero-shot ability as the original GPT3 on NLP tasks. But I won’t count this as a reason for rejection, as the authors have used reasonable methods and the unsatisfying results just show that this is an interesting and hard problem.

2. Hard to judge whether the results are meaningful or trivial. The main metric to judge the significance of the results is the difference between Frozen and Frozen_blind. However, as all numbers are so far away from a reasonable baseline model, it is hard to judge the meaningfulness of the difference between model numbers.
For example, on VQA, a blind Frozen (without seeing the image) gets 33.3 while a Frozen gets 38.2. The numbers are so far away from reasonable baselines (for example, on VQA, a language-only model without any visual information can get over 40% acc), it makes it hard to say whether the 5 points improvement is significant or not.

Detailed comments:
3. Line 126-127: it is stated that fine-tuning the LM hurts performance, which is somehow in contrast to what is observed in V&L model pre-training (e.g., ViLBERT, VisualBERT, LXMERT). I wonder what would be the possible reason.

4. Practically, if we just want a model with a good zero/few-shot ability on V&L tasks, we don't really need to fixate on keeping weights fixed during pre-training on multi-modal data? I see that the reason this paper keeps the weights frozen is simply because the performance is worse. I wonder if there are other reasons that we want to keep the weight frozen.

5. Frozen_train-blind: Table 1 says Frozen_train-blind is mentioned in Section 4.1, but I cannot seem to find it mentioned in the main text.

6. What is Frozen VQA? It says “mixing in data from VQA”. Does it mean mixing in VQA during pre-training?

7. Table 1: The Frozen-finetune is a bit confusing, as it could mean fine-tuning LM during Conceptual Caption pre-training or fine-tuning on VQA with few examples. Judging from the context, I assume it is the former but it would be helpful to be clearer on this.


**Time Spent Reviewing:**

2

---

> ### Author Response · Authors · 2021-08-11
> **Response to Reviewer DtyQ**
>
> Thank you very much for your review! Please see our top-level reply to all reviewers in the sibling thread where we have responded to your main points!
>
> Please also see below for answers to your specific questions.
>
> > Line 126-127: it is stated that fine-tuning the LM hurts performance, which is somehow in contrast to what is observed in V&L model pre-training (e.g., ViLBERT, VisualBERT, LXMERT). I wonder what the possible reason is.
>
> There are important differences between the models you mention here and the Frozen setting. In Frozen, the pre-training involves training on a large dataset of text alone, whereas in VilBERT etc. pre-training involves task-agnostic training on multimodal datasets (e.g. conceptual captions). VilBERT et al. are then followed by task-specific training, where the whole model is fine-tuned, whereas Frozen is next trained on a multimodal objective (captioning) - where we observe that freezing most of the model performs better than fine-tuning it all. Finally, Frozen is evaluated on new tasks in a few-shot manner (i.e. no further weight updates).
>
> > Practically, if we just want a model with a good zero/few-shot ability on V&L tasks, we don't really need to fixate on keeping weights fixed during pre-training on multi-modal data? I see that the reason this paper keeps the weights frozen is simply because the performance is worse. I wonder if there are other reasons that we want to keep the weight frozen.
>
> The value of freezing the language model in this process derives from the need in this particular setting for transfer between unimodal training and multimodal objective; while preserving the properties of the unimodal model (fast-binding, memory, linguistic and semantic knowledge). The key finding of this work is that, when we do this, the model can in addition adapt to further tasks without the need to train or fine-tune further. As you suggest, however, there are other pragmatic reasons why we may want to keep weights frozen. For instance, if one has a large trained language model, comparatively few additional parameters have to be trained to turn it into a multimodal model, which can save computation. It can also make it the challenge of building a multimodal system more manageable by breaking down the problem into components that different teams can specialize in.
>
> > Frozen_train-blind: Table 1 says Frozen_train-blind is mentioned in Section 4.1, but I cannot seem to find it mentioned in the main text.
>
> It is described in L191-192 - we will add a reference there back to the table to make this connection easier for the reader.
>
> > What is Frozen VQA? It says “mixing in data from VQA”. Does it mean mixing in VQA during pre-training?
>
> The multimodal training involves both captioning and answering questions from the VQAv2 training set. The training objective is the same in both cases; we just maximize the likelihood of Frozen generating the correct answer after first processing the image and the question. Because this model is explicitly trained on VQA, it is not doing few-shot learning of VQA, but we include it for comparison purposes.  When co-training on captioning and VQA, the VQA data accounts for 10% of training data due to its relatively smaller dataset size.
>
> > Table 1: The Frozen-finetune is a bit confusing, as it could mean fine-tuning LM during Conceptual Caption pre-training or fine-tuning on VQA with few examples. Judging from the context, I assume it is the former but it would be helpful to be clearer on this.
>
> Yes, that is correct. Thank you for the point, we will clarify in the script.

---

> > ### Comment · Reviewer_DtyQ · 2021-08-26
> > **Response**
> >
> > Thank you for the rebuttal. Now, most of my concerns are addressed.

---

> > > ### Author Response · Authors · 2021-09-02
> > > **Thank you.**
> > >
> > > Thank you, we are glad to hear that most of your concerns are addressed.

---

### Official Review · Reviewer_J1ff · 2021-07-18

**Rating:** 7
**Confidence:** 4

**Summary:**

The paper proposes a new way to reuse large pre-trained language models and condition their generation on visual input. The authors build on the idea of pretext tuning and make pretext a (CNN) function on an image. By training the CNN for VQA, keeping the language model decoder fixed, the authors bring visual information into the input space of the language model. The authors propose several benchmarks for few-shot classification and captioning and show that their model demonstrates promised capabilities and achieves better than random results.

**Ethics Review Area:**

["I don’t know"]

**Limitations And Societal Impact:**

-

**Main Review:**

While it was standard for the computer vision community to reuse large pre-trained CNNs for image/video or even multi-modal tasks, doing the same with language models was not possible. This paper proposes to reuse the large pre-trained language models in a simple and elegant way.
The paper is very well written and clearly conveys the main point. I like how the authors demonstrate numerous capabilities that are enabled by such design, in the Experiments. I believe that this work may pave the way for new interesting research in multi-modal learning, especially given that the authors introduce new benchmarks.

The only drawback I see in this paper is the weak results in the introduced benchmarks. While this is not critical at this stage, it would be useful to have a better idea why the results are low, i.e. is it the difficulty of the task or simply not very capable individual components that are comprising the pipeline.

**Time Spent Reviewing:**

3

---

> ### Author Response · Authors · 2021-08-11
> **Response to Reviewer J1ff**
>
> Thank you very much for the review! We are glad to hear that you found the paper valuable.
>
> Regarding your concerns about model performance, please see our top-level reply to all reviewers in the sibling thread.

---

### Official Review · Reviewer_GTTm · 2021-07-22

**Rating:** 8
**Confidence:** 4

**Summary:**

This paper proposes a way to train a few-shot image-text neural network in which the text encoder weights are frozen, but used to train the image encoder from scratch. It provides an interesting training framework and evaluates few shot performance on image classification and vqa tasks, also studying several qualities of interesting, e.g. rapid task adaptation, encyclopedic knowledge, and fast concept binding.

**Limitations And Societal Impact:**

1. There isn't analysis of samples that were already correctly classified with the blind models. What did those reveal, did you look at them? This would provide better insight to this issue and also help researchers build upon this work to specifically target these biases.
2. The classification baselines only explore at most a 5-way classification task, and with performance so low, it isn't particularly promising that results would extend well to a more realistic setting.
3. The role of the relative position encoding is not studied, and so it is unclear how necessary the image-text embedding ordering is for the experimental results.
4. The plateau or worsening of performance as the number of shots or repeats increases is not addressed, and is a bit concerning. Why would performance get worse with more examples?

**Main Review:**

The training paradigm is really cool, and definitely novel. The merit of this paper is strong as having few-shot learners that can adapt to tasks they've never seen before is very desirable. The paper is also generally well written and easy to follow. The primary drawback for the significance of this work is that the task performance is very low, but the authors do acknowledge that beating SOTA is not their intention. Also, the paper properly compares to "blind" baselines to prove that the network actually makes use of visual information, which is a necessary comparison and further verifies the merit of their work.

Other positives include that comparison to prior work is thorough and there are clear distinctions made between this papers contributions and those of related work.

Otherwise, there are some questions + comments:
- L34: what is meant by "ordered sets"? do you just mean pairs of images and words as input? it's a bit confusing.
- L73-80: the terminology of prefix vs. prompt vs. infix is not clear. Could you clarify the differences and revise the writing for clarity?
- L122: it seems like the term "n" is used to reference different concepts/quantities throughout the paper, which gets confusing. In L22, is the number of visual prefix tokens = n? this should be explicitly stated. Then, in tables 1,2 n refers to the number of shots. Then, L274 refers to n as the number of images of the referents of words.
- L130-138: is the difference between y and t that y are text tokens in their raw form, and t are their embeddings? from the notation it seems like they're the same thing, especially since x is defined on i_1, i_2
- L30-138: why is there swapping between i_1, i_2 and then generalizing it to i_1, i_2, ...., i_n?
- L136: can you define the relative position encoding? it's not defined/clear to me. Furthermore, did you ever actually check changing the image prefix positions? I would be curious to know how well the relative position encoding works, and how much the input ordering is needed as an inductive bias.
- L154: how do we know that the "different ways of inducing the task to the model" aren't potentially harmful or biased to different groups?
- Figure 4: The purpose of injecting nonsense words isn't explained early enough (it's only ever formally mentioned on L241-244), can you please address that sooner in the text. Also, why isn't there an answer in the upper row for the "Question from ImageNet"?
- Number of shots vs. number of inner shots. Are there two differences here? The first being that number of inner shots is per category, and the second being that the "full" example isn't provided for inner shots? I am trying to connect it to the VQA example in the number of shots definition, in which the answer is provided in addition to the image and question.
- Number of repeats. Repeating samples doesn't feel well motivated, and isn't really discussed in the text. Why is this done? I am not understanding how this helps us "explore how the model integrates visual information"
- L174: is the word "fast" supposed to precede general knowledge? what about this approach makes the access to general knowledge fast? or is the point that we don't have to retrain the language model to be task specific? It just seems a bit redundant with "rapid adaptation" and "fast binding"
- Table 1,2 Frozen_train-blind: why do these results do worse with n=4 shots?
- Table 1,2: clarifying question- are all of these results with having 2 visual prefixes?
- Table 1: How much VQAv2 data is mixed in?
- L189-191. This is an important take-away, but it is not new - see Visual Dialogue without Vision or Dialogue, Massiceti et al. Maybe you already are familiar with related work showing this, but nothing is cited here, so it would be good to mention this confirms results already shown in prior work.
- L207: Did you ever look at increasing the number of shots until full SGD training performance was reached? is it possible?
- Fast binding: could the purpose of fast binding be better motivated? right now the use of nonsense words doesn't really provide intuition of how this is useful. Do you think we could extend fast binding to other languages? That seems like it would be an impactful use case.
- Table 3: It seems like Frozen and Frozen (real name) plateau as both the number of inner shots or repeats increases, or performance even gets worse (repeats = 5, frozen real name). Why is this? This happens both for the 2-way and 5-way experiments.
- L241-242: What does "this allows the system to express its answers with word-like tokens" mean?
- L251-255 states that Frozen improves performance with more examples, which feels like an exaggeration of the results, given that the performance plateaus after 3 which is not addressed in the analysis.
- Fast-VQA and Real-Fast-VQA: Not enough dataset details were provided on this in the main text, additional details on the number of samples etc would be helpful. Also, I'd reconsider using "Fast" in the name, it's a bit confusing. And "Real-fast" sounds like the data is now really fast. Is the word fast being used because the data is evaluated in a few shot setting?

Style suggestions, typos, etc.
- L34: remove "e.g."
- L136: the ellipses here are an awkward way to end a sentence
- L149: people that aren't native English speakers may not know what a "stress test" is. I'd suggest rewording.
- L153: remove "at"
- L215: the e.g. feels awkward here, consider removing
- L258-259, 265: wording a bit weird/hard to follow
- L283: "a model is reminded what the important entities in the question look like" is a bit confusing- is this because the question now has the real category names included? I would reiterate that explicitly.
- L325-326: what does this mean?
- L327: adding a hyphen after language may help the flow here.

**Time Spent Reviewing:**

4 hours

---

> ### Author Response · Authors · 2021-08-11
> **Response to Reviewer GTTm**
>
> Thank you very much for your thorough review! It has helped us to make parts of the paper substantially clearer.
>
> Please see our top-level reply to all reviewers in the sibling thread where we have included responses to your main points.
>
> Please also see below answers to your specific questions. As for the stylistic and clarification comments, we will apply them in the final manuscript. Thank you again for those!
>
> > L34: what is meant by "ordered sets"? do you just mean pairs of images and words as input? It's a bit confusing.
>
> Yes, exactly. That is correct. We will update the manuscript to make it more clear.
>
> > L73-80: the terminology of prefix vs. prompt vs. infix is not clear. Could you clarify the differences and revise the writing for clarity?
>
> At a high level:
> - 'prompt': something that the model conditions on that itself is conditioned on input data (either words or images)
> - 'prefix': additional weights that are not conditioned on input, but could be conditioned on e.g. task type (as in 'prefix tuning')
> We will clarify this and make our usage of the terms more consistent throughout the paper.
>
> > L30-138: why is there swapping between i_1, i_2 and then generalizing it to i_1, i_2, ...., i_n?
>
> The motivation for using the general notation is that one can potentially use more than 2 or fewer than 2  “image tokens” to represent each image. We experimented with this and found that 2 performs best; but we intend to continue investigating this in future work.
>
> > L154: how do we know that the "different ways of inducing the task to the model" aren't potentially harmful or biased to different groups?
>
> This is an important point. In the current system, researchers or users are free to choose the task induction string, and there is no scope here for an exhaustive study on the potential biases or other effects introduced by this. There is excellent research on mitigating bias in the deployment of language models, which we will add reference to when describing this aspect of the model.
>
> > Number of shots vs. number of inner shots. Are there two differences here? The first being that the number of inner shots is per category, and the second being that the "full" example isn't provided for inner shots? I am trying to connect it to the VQA example in the number of shots definition, in which the answer is provided in addition to the image and question.
>
> Yes, that is correct. We consider a shot to be when we show the model how to complete the task, i.e. we show it a full example of a task. For VQA, an image + question + answer is one shot.
>
> > Number of repeats. Repeating samples doesn't feel well motivated, and isn't really discussed in the text. Why is this done? I am not understanding how this helps us "explore how the model integrates visual information".
>
> If the model performs better with more 'shots' (different instances of the same image category) than with more 'repeats' (multiple instances of the same image), this suggests that the model benefits from additional information and evidence when learning to bind a word to an image class. If there was no difference between shots and repeats, it would tell us that it is simply repetition of the same information that reinforces the meaning of the new word in the model. Because in general on open-ended miniImagenet we observe that providing more 'shots' is more effective than providing more repeats, this is some evidence that the model is indeed exploiting some of the diversity of the available visual information to form a general sense of the category that a new word corresponds to.
>
>
> > Table 1,2: clarifying question- are all of these results having 2 visual prefixes?
>
> Yes, across the paper, we are using a single model with 2 visual prefix tokens.
>
> > L207: Did you ever look at increasing the number of shots until full SGD training performance was reached? Is it possible?
>
> Yes, we did explore increasing the number of shots further, however as discussed there seems to be a limit to how long sequences of images and text Frozen can handle, since it’s trained on only inputs with a single image. We have added a clear statement of this fact into the paper.
> But indeed, there is room for improvement, and we and other groups are already thinking how to do better.
>
> > L241-242: What does "this allows the system to express its answers with word-like tokens" mean?
>
> We will clarify this in the script. We wanted to motivate using nonsense words versus the original miniImagenet labels, which are integers 1, 2, and so on. We wanted to convey the idea that the language-model should be more used to responding with words than with numbers. However, the main motivation for doing this is that the integers 1, 2, and so on may already 'mean' something to the underlying language model (it will have experience of their natural order), which could bias its behavior more than using nonsense words which should have little or no prior meaning to the language model.
>
> > Table 1: How much VQAv2 data is mixed in?
>
> 20%. We will add this to the manuscript.
>
> > Fast binding: could the purpose of fast binding be better motivated? right now the use of nonsense words doesn't really provide intuition of how this is useful. Do you think we could extend fast binding to other languages? That seems like it would be an impactful use case.
>
> Our goal is to understand whether the model can learn the meaning of a new word quickly, which is potentially useful for all sorts of applications: adapting to new users and their chosen style of language, giving the sense that a user can 'teach' the model, and as you suggest there could even be applications where a model could quickly learn a new language!
>
>
> The motivation for using nonsense words in our controlled experiments was to make sure that the pre-trained LM does not know them beforehand and cannot connect it to the image classes.
> Using nonsense words may feel contrived, but -- for the reason above -- it is a standard way to test whether human learners can learn the meaning of a new word quickly (the word 'dax' was introduced in such experiments on humans). So we think it is reasonable to apply this established and well-studied method of measuring fast word-learning to machines.
>
> > Fast-VQA and Real-Fast-VQA: Not enough dataset details were provided on this in the main text, additional details on the number of samples etc would be helpful. Also, I'd reconsider using "Fast" in the name, it's a bit confusing. And "Real-fast" sounds like the data is now really fast. Is the word fast being used because the data is evaluated in a few shot settings?
>
> We have provided more details on the datasets in the Appendix, and we are working on open-sourcing them. We will change the name to "Guided VQA" as we agree the current name is imperfect.
>
> > L325-326: what does this mean?
>
> A potential application of Frozen would be that of, in the case of a malicious event where someone screens for some property in images on the web using image-language models, a personal assistant can screen your images to alert you as well. So even though individuals may not have power or ability to screen the whole web, they can still have personal mitigation. We will revise the manuscript to clarify that.
>
> > The role of the relative position encoding is not studied, and so it is unclear how necessary the image-text embedding ordering is for the experimental results.
>
> Attention over a sequence of keys/values is invariant to their position. The standard way to allow attention to choose where to look based on position is to add absolute position encodings expressed as learned one-hot embeddings or fixed sinusoidal absolute position representations. We instead use the method where the encoded position index is **relative** to the query position instead of absolute. Intuitively, we believe that relative position encodings are more appropriate for the case of images (where each image is represented by multiple contiguous tokens). We will add a paragraph in the Appendix explaining this in more detail. At this point we do not have a controlled experiment at this point comparing them head to head since it would require training a comparable large language model.

---

> > ### Comment · Reviewer_GTTm · 2021-08-23
> > **Rebuttal**
> >
> > Thank you for your thorough response. I think including these additional explanations into the main paper will help greatly!

---

> > > ### Author Response · Authors · 2021-09-02
> > > **Thank you.**
> > >
> > > We agree and thank you for your thoughtful discussion.

---

### Author Response · Authors · 2021-08-11
**Response to common comments**

In this thread, we would like to follow up on a few common points raised by the reviewers.

### Frozen performance vs. SOTA
We acknowledge the concern of reviewers that Frozen’s performance is a lot lower than other models on the established tasks that we consider. As we try to emphasize in the paper, comparison between Frozen and more specialized models can be a bit misleading:(a) Frozen is not directly trained on any of the evaluation tasks (unlike SOTA on those tasks), instead it rapidly learns only by being presented with a handful of samples of the task during evaluation, and (b) Frozen is an open-ended system capable of producing arbitrary language output (open evaluation), whereas SOTA systems on vision and language tasks are typically evaluated by ranking a fixed set of possible answers (closed evaluation). For the setting in which Frozen is evaluated, we believe the fact the system is capable of any few shot learning at all (by outperforming strong baselines in a few-shot setting on VQA-style tasks and significantly above-chance on meta-learning benchmarks like miniImagenet) is a notable result worth sharing with the community. We think that reporting these results will stimulate other researchers to improve this and related methods, a path that may ultimately yield flexible and more performant multimodal language technologies.

###  Frozen performance with increasing number of shots
Reviewers also raised a question about Frozen’s performance plateauing with increasing number of shots. We believe reason for this is that, when training, Frozen is only trained to caption a single image; the more images that the model is presented with at test time, the further this input is from the training data/distribution, both in terms of the number of images and where in the sequence of inputs they appear. Thus at test time there seems to be a trade-off between the model benefitting from more useful context and information with which to adapt to the task at hand, and uncertainty at dealing with an input of unfamiliar format. We add a discussion of this possibility to the results in the paper, as well as signaling the potential for addressing this limitation via smarter training regimes in future work.

### Summary of new appendix comparing blind and sighted models
Reviewers asked for a qualitative demonstration of Frozen responding to the contents of the image and thereby improving performance over the blind models on the VQAv2 dataset. We agree that this would be a valuable addition to the appendix and here summarize the results that will be described therein. We emphasize that in our view the rest of our results in the paper demonstrate clearly that Frozen refers to the contents of the image -- as an extreme example consider the miniscule probability that a language model would assign to the string _"The Wright Brothers"_ in the absence of context about airplanes (which it only gets from the image). Results on the open-ended Mini Imagenet evaluation also reassure us in this regard -- in that task there is no way a 'blind' model can exploit the language signal to achieve the above-chance performance that we see from Frozen.

**Specific examples from VQAv2:**

Frozen unsurprisingly outperforms the baseline on questions such as _"What animal is pictured?"_ (photo of giraffe) - Frozen responds with _"a giraffe"_ whereas the blind equivalent guesses _"monkey"_.

If the question is a bit more specific; (_"What are the sheep eating?"_), the blind model can correctly infer _"grass"_ - whereas Frozen produces _"they are eating the grass"_, which does not match the correct answer after stemming and normalization, so Frozen loses a point relative to the baseline.

For a picture of a person surfing and the question _"What is the man doing?"_, the blind baseline responds with _"he is trying to get a better view of the woman"_ (incorrect) whereas Frozen correctly answers _"he is surfing a wave"_ - but again receives a score of zero because the correct answer is simply _"surfing"_.

These final examples show the extent to which the answer-matching metric can influence the performance, and why performance for an open-ended system like Frozen can be a lot lower than for classification-based approaches to VQA.

---

> ### Comment · Reviewer_DtyQ · 2021-08-26
> **Response**
>
> Reading the specific examples from VQAv2, I think one way to address the variance resulted from "open-ended" evaluation is to adopt the common practice in VQA and treat it as a multi-class classification problem. VQAv2 has a few thousand candidate answers. Thus, one way to "generate" the answer is to evaluate the few thousand candidate answers against Frozen (calculate the probability that Frozen generates each candidate answer given the question & visual context and pick the answer with the highest probability (perhaps normalize it by length)). Would that give a more accurate assessment of the models?

---

> > ### Author Response · Authors · 2021-09-02
> > **Regarding open vs closed ended evaluation**
> >
> > Although we agree that doing "close-ended" evaluation could reduce the variance, we don’t think it would **necessarily** give a more accurate assessment of the models. It can reward models that are trained as classifiers (using the same set of possible answers as targets) - even though they would necessarily fail on any similar task whose answers were outside of that set. It would also make scores dependent on the particular choice of answer set, and can lead to later confusion about which set was used. Instead, we’d like to move towards more flexible generative models of language, and evaluations (even if imperfect) that reflect the way in which these models are applied - namely considering full sequences sampled from the model.

---

### Decision · Program_Chairs · 2021-09-27

**Decision:**

Accept (Poster)

**Comment:**

This work presents a proof-of-concept for multimodal few-shot training of frozen language models — where the parameters of a pre-trained language model are kept fixed while an image encoder is trained to prompt it to generate text. Despite its simplicity, the approach demonstrates surprising generalization capabilities of the model when presented unseen visual concepts, although as pointed out by the authors themselves, the results are still far from what current models can achieve on these tasks and more work is needed to refine these ideas further.